

# The semantic basis of taste-shape associations

Carlos Velasco[1,2], Andy T. Woods[1,3], Lawrence E. Marks[4,5], Adrian David Cheok[2,6] and Charles Spence[1]

[1] Crossmodal Research Laboratory, Department of Experimental Psychology, University of Oxford, Oxford, UK
[2] Imagineering Institute, Iskandar, Malaysia
[3] Xperiment, UK
[4] Sensory Information Processing, John B. Pierce Laboratory, New Haven, CT, USA
[5] School of Public Health and Department of Psychology, Yale University, New Haven, CT, USA
[6] School of Mathematics, Engineering, and Computer Science, City University, London, UK

Corresponding author
Carlos Velasco,
carlos@imagineeringinstitute.org

## ABSTRACT

Previous research shows that people systematically match tastes with shapes. Here, we assess the extent to which matched taste and shape stimuli share a common semantic space and whether semantically congruent versus incongruent taste/shape associations can influence the speed with which people respond to both shapes and taste words. In Experiment 1, semantic differentiation was used to assess the semantic space of both taste words and shapes. The results suggest a common semantic space containing two principal components (seemingly, intensity and hedonics) and two principal clusters, one including round shapes and the taste word "sweet," and the other including angular shapes and the taste words "salty," "sour," and "bitter." The former cluster appears more positively-valenced whilst less potent than the latter. In Experiment 2, two speeded classification tasks assessed whether congruent versus incongruent mappings of stimuli and responses (e.g., sweet with round versus sweet with angular) would influence the speed of participants' responding, to both shapes and taste words. The results revealed an overall effect of congruence with congruent trials yielding faster responses than their incongruent counterparts. These results are consistent with previous evidence suggesting a close relation (or crossmodal correspondence) between tastes and shape curvature that may derive from common semantic coding, perhaps along the intensity and hedonic dimensions.

"She laughed, a *laugh sweeter* than *honey*, with a *sound curving* and *zigzagging*, as if singing"

(Mo Yan, *Ball-Shaped Lightning*, cited by Yu, 2003, p. 190)

## INTRODUCTION

Several studies show that people systematically match both basic taste words and tastants (i.e., chemicals that generate gustatory sensations) with shapes that vary in terms of

[1]Here, we are mainly interested in the way in which taste/shape correspondences occur in the general population. As a consequence, research on synaesthesia, a rare condition where the stimulation of one sensory modality leads to concurrent sensory experiences in the same or other modalities, falls out of the scope of the present study. It is important to mention, though, that cases of taste-shape synaesthesia have been reported elsewhere (*Cytowic, 1993*; *Cytowic & Wood, 1982*).

their curvature (see *Cytowic & Wood, 1982*; *Spence & Deroy, 2014*; *Spence & Ngo, 2012*, for reviews). Over the last few years, researchers, including ourselves, have studied crossmodal (taste-shape) correspondences,[1] providing some hints as to their underlying mechanisms (e.g., *Velasco et al., 2015a*; *Velasco et al., 2016*) and their effects on taste (or gustatory) information processing more generally (e.g., *Gal, Wheeler & Shiv, 2007*; *Liang et al., 2013*). Notably, while the way in which people match basic tastes with shapes seems reasonably well understood, the mechanisms that underlie crossmodal correspondences, as revealed in crossmodal (taste-shape) matches and congruency effects in perceptual processing are still to be clarified. In particular, research still needs to clarify when, how, and why the mechanism(s) that underlies taste-shape correspondences may influence the processing of taste (perceptual and linguistic) and shape information.

Velasco and his colleagues have investigated how people match basic tastes and shapes. So, for example, the results of one series of four experiments revealed that people associate sweet (both when presented as a word and as a tastant) with round shapes, and bitter, salty, and sour (as words and tastants) with more angular shapes (*Velasco et al., 2015a*, see also *Ngo et al., 2013*; *Velasco et al., 2014*; *Velasco et al., 2015b*; *Wan et al., 2014*). What is more, *Velasco et al. (2015a)* also reported that the more the participants liked the taste (but not a taste word), the rounder the shape matched to it (see also *Bar & Neta, 2006*, on curved objects preference) and suggested a hedonic mechanism to explain the crossmodal matching (see also *Ghoshal, Boatwright & Malika, 2015*). This finding was subsequently replicated by *Velasco et al. (2016)*. The latter researchers found that taste concentration can also affect shape matching, with more versus less intense tastants more likely matched to angular versus round shapes, respectively (see also *Becker et al., 2011*). Given the focus of the present study—on the semantic basis of taste word/shape correspondences, and associated congruence effects—the aforementioned findings are intriguing. Nevertheless, correlations alone do not suffice to show that a hedonic mechanism underpins the correspondences. Moreover, it is important to evaluate a wider range of intensities, given that the authors tested just two concentrations.

The idea that certain crossmodal correspondences may be mediated by the affective properties of the matching or mismatching stimuli is certainly not new (e.g., *Kenneth, 1923*; see also *Marks, 1996*, for a review). Indeed, researchers have demonstrated recently that the way in which people match music and colours (*Palmer, Langlois & Schloss, 2015*; *Palmer et al., 2013*) and colours and fragrances (*Guerdoux, Trouillet & Brouillet, 2014*; *Schifferstein & Tanudjaja, 2004*) can be mediated by emotion. In an earlier study, *Collier (1996)* demonstrated that judgments of many sets of visual and auditory stimuli could be reduced to two dimensions, identified, by the author, as valence and activity. Further, some of the visual and auditory stimuli overlapped on these dimensions. Collier's generalization is relevant to taste/shape correspondences, which also might overlap on the same affective dimensions. What is more, such an idea is important in the context of multisensory perception in general, given that, in addition to factors such as spatiotemporal alignment and semantic congruence (*Stein, 2012*), crossmodal correspondences can also help mediate multisensory perception (e.g., *Parise, Knorre &*

*Ernst, 2014*). Indeed, crossmodal correspondences can provide relevant information to people when they make inferences about the (often) noisy sensory world (*Parise, 2015*).

Importantly, earlier research also points to the notion that taste/shape correspondences may influence the processing of taste-information. For instance, *Liang et al. (2013)* assessed the influence of shapes on people's sensitivity to sweetness using near-threshold sucrose solutions. In their study, people rated round shapes as more pleasant. Further, presenting a round shape rather than an angular shape before tasting a sweet solution enhanced sweetness sensitivity. Unfortunately, however, this study is the only of its kind, and replication is critical (the effect is certainly specific and small), perhaps using everyday, suprathreshold, solutions. Moreover, there is a possible confound of response bias in the study, as *Liang et al. (2013)* did not attempt to control for the participants' response criterion (e.g., apparently they did not include any 'blank,' water trials).

*Gal, Wheeler & Shiv (2007)*, reported a study in which their participants were asked to indicate which of three shapes (which could be all round or angular) had the largest surface area before rating a piece of cheddar cheese. The results showed that the curvature of the shapes presented in the first task influenced the perceived sharpness of the cheese, with the angular shapes leading to higher sharpness ratings than the round shapes. What is more, other studies have shown that the shape of a plate and food (when it is round as compared to angular) can influence participants' sweetness ratings of the food (resulting in people rating the food as tasting sweeter, see *Fairhurst et al., 2015*; *Stewart & Goss, 2013*; see also *Piqueras-Fiszman et al., 2012*).

Here it is worth mentioning that, in spite of their perceptual basis, similarities across the senses also surface in language (e.g., see the quote at the beginning of the Introduction, *Marks, 1978*; *Marks, 1996*). With this in mind, we ask whether the potential hedonic- and intensity-related explanations/mediations of taste/shape correspondences may extend to taste words and, if so, whether they reflect: a perceptual process; a common connotative meaning (*Walker, 2012*; *Walker, Walker & Francis, 2013*; see also *Collier, 1996*; *Karwoski, Odbert & Osgood, 1942*, for earlier examples); or perhaps, a combination of the two (see also *Walker & Walker, in press*). According to the semantic coding hypothesis (SCH, see *Martino & Marks, 2001*), high level mechanisms that connect information across the senses may emerge from developmental experiences with various percepts that are coded into language, and that can affect multiple levels of human information processing. Consequently, crossmodal congruence effects can arise not only in the processing of perceptual stimuli, but also in the processing of verbal stimuli (*Martino & Marks, 1999*). While *Liang et al. (2013)* provided some evidence that congruent shapes can influence people's detection of sweet solutions presented at near threshold levels, we ask here whether the congruence of taste words and shapes can affect perceptual processing. As *Marks (1978)* pointed out, "According to the Oxford English Dictionary, 'sharp' applied first to touch, then subsequently to taste (ca. 1,000), visual shape (1,340), and hearing" (p. 190), indicating that shape-related words have been used to describe tastes for several centuries, and thereby perhaps some kind of implicit relation between shape and taste quality (see also *Williams, 1976*; *Yu, 2003*).

Here, we describe two experiments designed to assess whether shapes and taste words share a common semantic space and whether congruence between them can influence both taste words and shape information processing. Experiment 1 used semantic differentiation (*Osgood, Suci & Tannenbaum, 1957*) to assess whether taste words and shapes share common dimensions of connotative meaning. Experiment 2 used a speeded classification task to assess whether taste/shape congruence affects the categorization of taste words and shapes. We hypothesized that taste words and shapes share a common semantic space to which previously reported associations will project, and that people will respond faster to the congruent versus incongruent pairings.

## EXPERIMENT 1

### Methods and materials

#### Participants

A total of 102 participants (M age = 34.7 years, SD = 11.8, age range = 19–70, 51 females) took part in the study, online through the Adobe Flash based Xperiment software (http://www.xperiment.mobi). The participants were recruited using Amazon's Mechanical Turk in exchange for a payment of 1.50 USD (see *Woods et al., 2015*, for a methodological overview of internet-based research). All of the participants were based in the USA, and all agreed to take part in the study after reading a standard consent form. The experiment was reviewed and approved by the Central University Research Ethics Committee at the University of Oxford (MS-IDREC-C1-2014-056).

#### Apparatus and materials

The images of four shapes (previously used by *Köhler, 1929*; *Ramachandran & Hubbard, 2001*), two angular and two round (see Fig. 1), as well as four taste words, namely bitter, sour, salty, and sweet, were the stimuli in this study. The taste words were presented in font Times New Roman 80.

Each stimulus was assessed using the semantic differential technique (SDT). This is a procedure in which the connotative meaning of objects and concepts is measured by using rating scales with polar adjectives (see the original work of *Osgood, Suci & Tannenbaum, 1957*, for details; see also *Albertazzi et al., 2014*, for a recent example). Twelve pairs of polar adjectives were included, which were based on previous research using the SDT (*Osgood, Suci & Tannenbaum, 1957*; *Osgood, 1964*). Each pair has been found to correlate with three bipolar dimensions, namely, evaluation, potency, and activity. The pairs of adjectives were: (1) nice–awful, (2) good–bad, (3) mild–harsh, (4) happy–sad (evaluation), (5) powerless–powerful, (6) weak–strong, (7) light–heavy, (8) shallow–deep (potency), (9) slow–fast, (10) quiet–noisy, (11) passive–active, and (12) dead–alive (activity). Each shape and taste stimulus was rated on a 100-point visual analogue scale (VAS), unmarked except for the adjectives, located outside the poles of the scale. Adjectives within the pairs 1, 3, 6, 8, 10, and 12 were reversed during in the experiment.

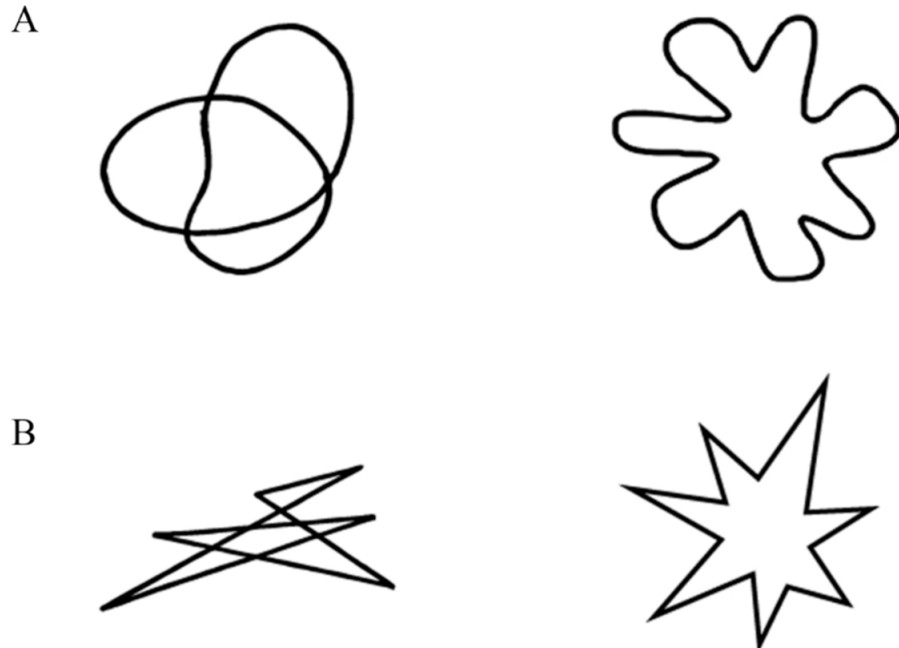

**Figure 1** Shape stimuli used in Experiment 1.

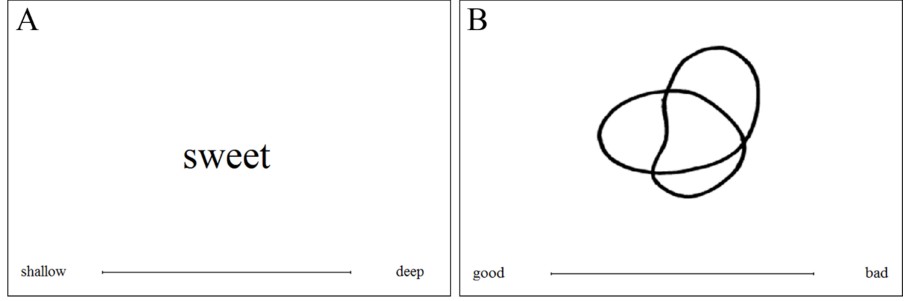

**Figure 2** Example of (A) A taste word and (B) A shape trial in Experiment 1.

### Procedure

At the beginning of the study, all participants were informed about the general aims and agreed to take part after reading a standard consent form. In the instructions, the participants were told that they would be presented with taste words or shapes and asked to rate them on a number of different VAS scales. On each trial, one of the stimuli (shape or taste word) was presented in the middle of the screen together with a VAS (see the example in Fig. 2). Trials were blocked by pair of adjectives (scales), and both order of trials and order of blocks were randomized across participants. In each block of adjective-defined scales, participants responded to the eight stimuli (four shapes and four taste words), giving rise to a total of 96 trials.

**Table 1** Varimax-rotated component matrix in Experiment 1 (see also Fig. 4).

| Adjectives | Component | |
|---|---|---|
| | 1 | 2 |
| Passive–active | **.995** | −.018 |
| Slow–fast | **.986** | .033 |
| Powerless–powerful | **.986** | −.065 |
| Weak–strong | **.980** | −.125 |
| Shallow–deep | **.934** | −.182 |
| Quiet–noisy | **.933** | −.184 |
| Harsh–mild | **−.825** | .565 |
| Sad–happy | .168 | **.979** |
| Awful–nice | −.293 | **.940** |
| Bad–good | −.311 | **.919** |
| Light–heavy | .497 | **−.825** |
| Dead–alive | .560 | **.808** |
| Eigenvalues | 7.09 | 4.43 |
| % of variance | 59.09% | 36.89% |

*Analysis*

A varimax-rotated principal component analysis (PCA) was used to define the principal dimensions arising from the different scale ratings of tastes and shapes. In addition, a hierarchical cluster analysis with Ward's method and squared Euclidean distance as the similarity measure (see *Kaufman & Rousseeuw (2005)*, for more details) was conducted in order to assess whether the different tastes and shapes would group as a function of common ratings in the scales used in Experiment 1. The aforesaid analyses were performed with IBM SPSS v. 19 and the PCA visualizations were created in the R′ (*R Core Team, 2015*) {FactoMineR} package (see http://factominer.free.fr/). The data were aggregated as a function of dimensions and clusters and Wilcoxon signed-rank tests were performed in R, to assess any difference between clusters as a function of dimensions. Effect sizes were calculated by means of Cliff's Delta as implemented in the {effsize} package in R (see https://cran.r-project.org/web/packages/effsize/effsize.pdf), in which 0 indicates the absence of an effect (the distributions overlap), while a value of −1 or 1 indicates a large effect (no overlap whatsoever; see *Cliff, 1996*).

**Results and discussion**

The principal component analysis (PCA, see Fig. 3) revealed that two components had eigenvalues over Kaiser's criterion of 1 and, in combination, explained 95.98% of the variance. Table 1 shows the factor loadings after the varimax (orthogonal) rotation. Note that the first and second components accounted for 59.09% and 36.89% of the variance, respectively.

The dendrogram resulting from the hierarchical cluster analysis appears in Fig. 4. Two major clusters are evident (see Table 2), one grouping round shapes with the taste word "sweet," and another grouping angular shapes with the taste words "salty," "sour," and
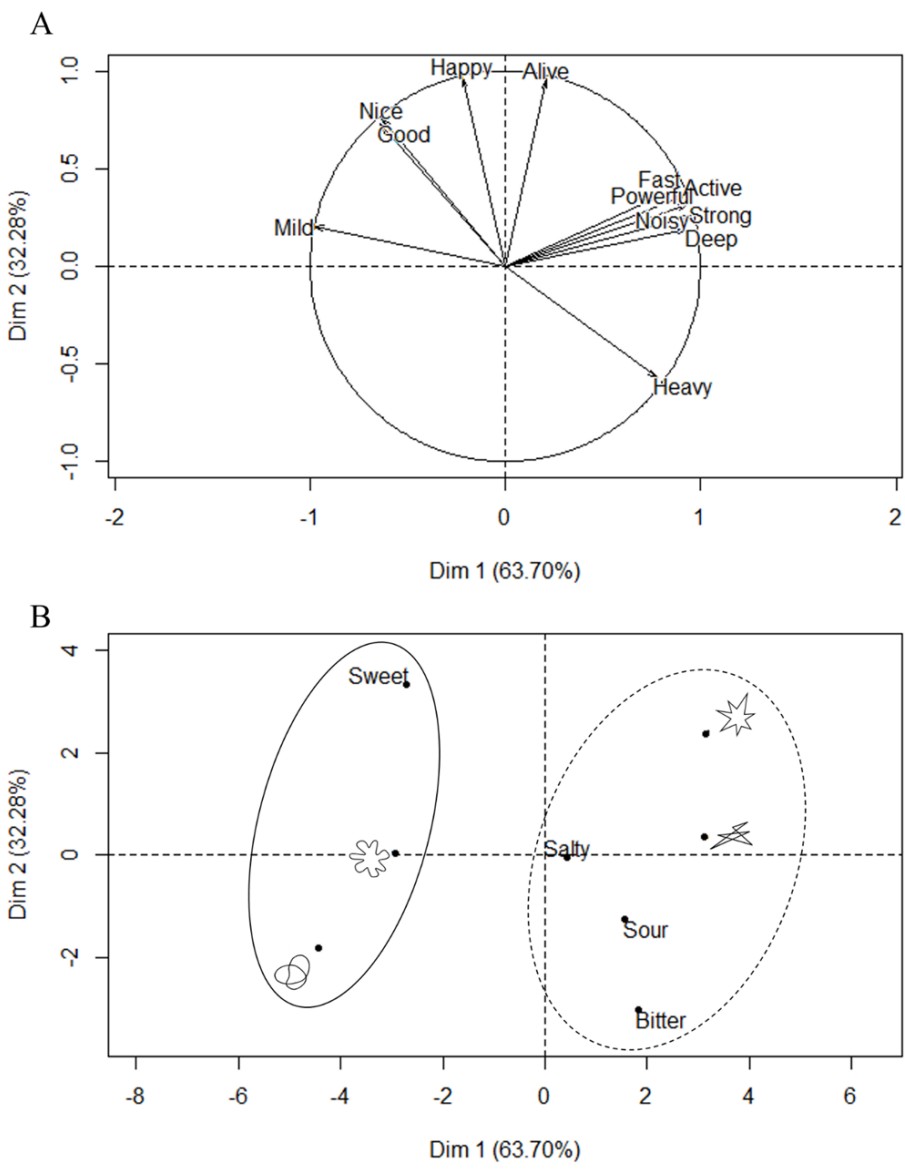

**Figure 3** **(A) Unrotated factor map of the polar scales in Experiment 1. Note that only the label of the upper end of the scales is presented. (B) Unrotated factor map for the stimuli.** The circles grouped the variables as a function of the two clusters identified in the subsequent cluster analysis. Note that given that (A) and (B) show the unrotated visualizations, the percentages for each component vary slightly from those presented in Table 1.

"bitter." These groupings reflect the tendency for stimuli in each cluster to receive similar ratings on the different semantic differential scales.

After identifying the two principal components and the two clusters, the data were aggregated as a function of dimension and cluster (Fig. 5 summarizes the mean values). Note that the scores of harsh/mild and light/heavy were reversed as they correlated negatively with their respective dimensions. Wilcoxon signed-rank tests were performed in order to assess any difference between clusters on each dimension. The ratings on the

**Table 2 Hierarchical cluster analysis in Experiment 1.** Note that the distance measure was rescaled to a ~0–1 range, and that the first large jump occurs between stages four and five.

| Stage | Cluster combined | | Coefficients | Stage cluster first appears | | Next stage |
|---|---|---|---|---|---|---|
| | Cluster 1 | Cluster 2 | | Cluster 1 | Cluster 2 | |
| 1 | Sour | Bitter | .000 | 0 | 0 | 4 |
| 2 | [shape] | [shape] | .010 | 0 | 0 | 6 |
| 3 | [shape] | [shape] | .031 | 0 | 0 | 5 |
| 4 | Salty | Sour | .072 | 0 | 1 | 6 |
| 5 | [shape] | Sweet | .220 | 3 | 0 | 7 |
| 6 | [shape] | Salty | .400 | 2 | 4 | 7 |
| 7 | [shape] | [shape] | 1.155 | 6 | 5 | 0 |

first dimension of the stimuli in the second cluster were higher than those in the first cluster ($p < .001$, Cliff's Delta $= 0.96$), whereas the ratings on the second dimension of the stimuli in the first cluster were lower than those in the second cluster ($p < .001$, Cliff's Delta $= 0.79$). In other words, the round shapes and the taste word "sweet" were rated as more positively-valenced and less intense than the angular shapes and the taste words "salty," "sour," and "bitter."

These results provide further support for the presence of an association between the word "sweet" and round shapes and the words "bitter," "salty," and "sour" and angular shapes (*Velasco et al., 2015a*; *Velasco et al., 2015b*; *Velasco et al., 2016*). Moreover, the results also suggest that tastes and shapes share a semantic space, or a set of implicit meanings, which is initially characterized by two main components. Indeed, a possibility is that these components reflect the two elements identified by *Velasco et al. (2015a)* and *Velasco et al. (2016)*, namely hedonic value and intensity. Consistently, the results of Experiment 1 fall in line with the idea that perceptual dimensions (e.g., sweet vs. sour, and fast vs. slow) differentiate between valence and arousal in specific ways (e.g., positive and negative and high and low arousal, respectively, see *Cavanaugh, MacInnis & Weiss, 2015*). Two limitations may be mentioned in regard to this study. First, it is possible that blocking by pairs of adjectives may have increased contrast between stimuli in each dimension. Second, the number of shapes included is very limited. This said, the common semantic space between round shapes and angular shapes, as well as their differences, are certainly consistent with previous research using those shapes (*Holland & Wertheimer, 1964*; *Lindauer, 1990*; *Lyman, 1979*).

Experiment 1 provides evidence in support of the idea that taste words and visual shapes share dimensions of connotative meaning. Given this, Experiment 2 aimed to assess whether the crossmodal correspondence between taste words and shapes would produce congruence effects over-and-above those already reported with tastes per se
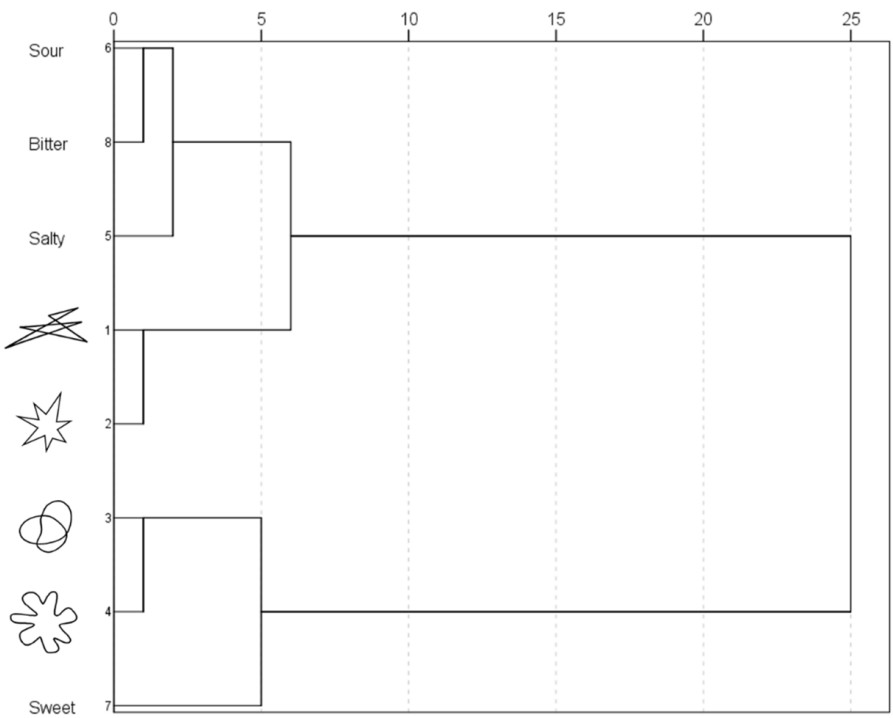

**Figure 4** Dendrogram obtained by means of hierarchical cluster analysis in Experiment 1.

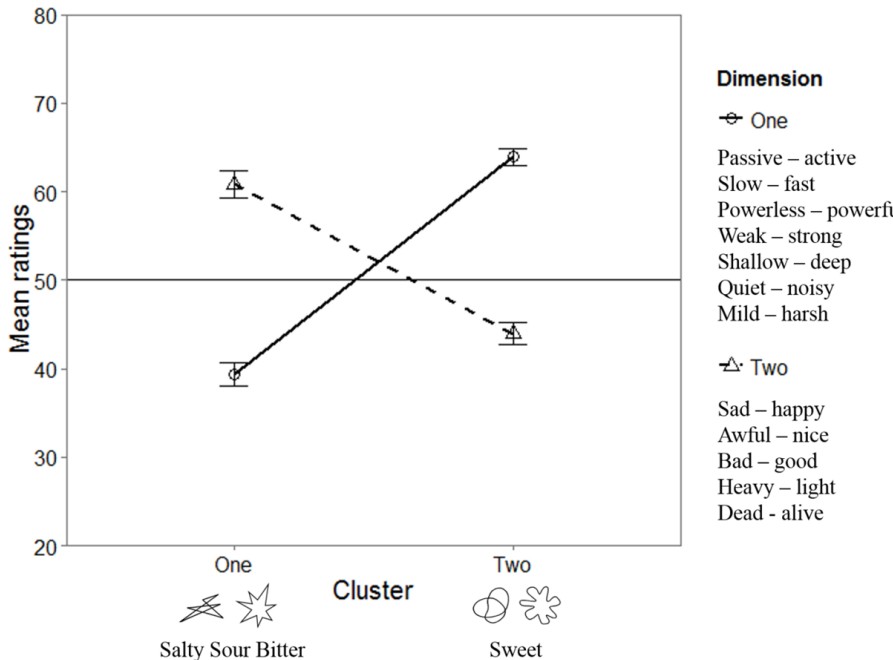

**Figure 5** Mean ratings for each cluster and dimension in Experiment 1. The error bars represent the standard error of the means.

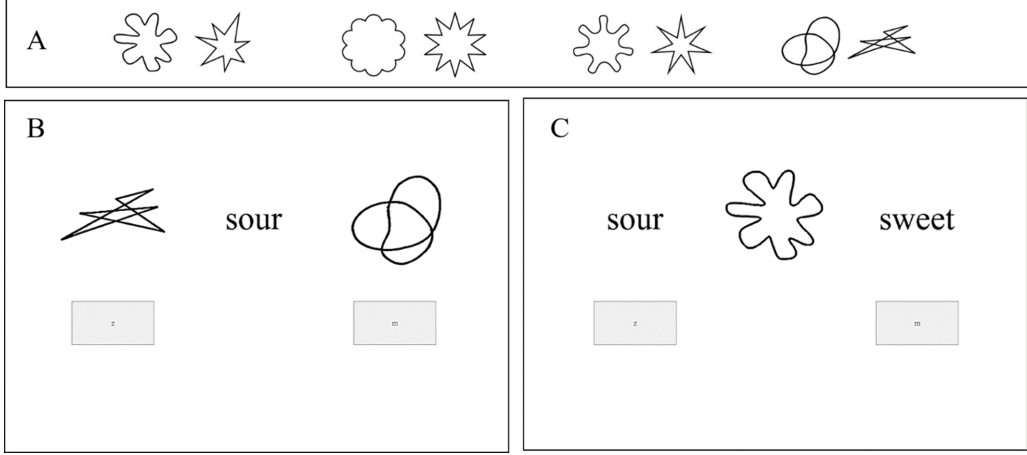

**Figure 6** **(A) The shape stimuli used in both tasks, (B) A trial in the shape response task, and (C) A trial in the taste response task.** Note that the shape stimuli are group in pairs as used as responses for the shape response task.

(*Liang et al., 2013*), that is, by using linguistic taste stimuli. For this purpose, a task was designed in which a larger sample of participants (in order to compensate for potential hardware-related differences across participants and fewer trials, e.g., *Woods et al., 2015*) were given congruent or incongruent instructions about the mapping between taste words and shapes and were later asked to respond to shapes or taste words with taste words and shapes, respectively.

## EXPERIMENT 2

### Methods and materials

#### Participants

A total of 253 participants (M age = 34.48 years, SD = 10.90, age range = 18–73 years, 138 females) took part in the study online and received a payment of 1.80 USD. All were based in the USA, and all agreed to take part in the study after reading a standard consent form.

#### Apparatus and materials

The ten stimuli comprised four pairs of shapes (one round and one angular within each pair, 200 × 200 pixels each; see Fig. 6A), plus two taste words "sweet" and "sour." The taste words were again presented in font Times New Roman 80.

#### Procedure

The participants took part in two tasks. In one of the tasks (*shape response*, see Fig. 6A), the participants were presented the taste words "sweet" or "sour" (one at a time) and asked to respond with either an angular or round shape (i.e., pairs of shapes taken from Experiment 1, see Fig. 6B). In the other task (*taste response*, see Fig. 6C), the participants were presented the eight shape stimuli (one at a time) and were asked to respond with the taste words "sweet" or "sour." Taste/shape congruence was manipulated in both tasks.

**Table 3  Experimental design used in Experiment 2.**

| Task | Congruence | Instructions (stimuli mapping) | Stimuli | Responses | Unique trials | Repetitions |
|---|---|---|---|---|---|---|
| Shape response | Congruent | Sweet–round and sour–angular | Taste words (sweet or sour) | Angular or round shape (four pairs ×2) | 8 | |
| | Incongruent | Sweet–angular and sour–round | | | 8 | X2 |
| Taste response | Congruent | Round–sweet and angular–sour | Shapes (eight shapes) | Sweet or sour | 8 | |
| | Incongruent | Round–sour and angular–sweet | | | 8 | |

That is, each task included a block of congruent trials and a block of incongruent trials. Both task order and block order were randomized across participants. In the congruent (incongruent) block of the *taste response task*, the participants were asked to respond with the word sweet every time they saw a round (angular) shape and with the word sour every time they saw an angular (round) shape. In the congruent (incongruent) block of the *shape response task*, the participants were instructed to respond with round shapes every time they saw the word sweet (sour), and with angular shapes every time they saw the word sour (sweet). Note, however, that the possible responses were presented to the left or to the right of the target stimulus, and the participants would have to press z or m, a function of the position of the correct response (see below). In both tasks, the participants were instructed to respond as rapidly and accurately as possible to a target stimulus (taste words or shapes) by pressing the key that was associated the stimulus that matched the parings in the instruction (shapes and taste words, respectively).

Table 3 summarizes the experimental design. Each of the tasks included eight unique trials. In the *shape response task*, half of the trials required of the participants to respond to the word "sweet" and the other half to the word "sour." Moreover, four trials included the round shapes on the right and the angular on the left (two for "sweet" and two for "sour"); in the remaining four trials, the positioning was reversed. In the *taste response task*, the participants responded to the eight shapes with words "sweet" and "sour." The right-left position of "sweet" and "sour" was thus fully counterbalanced.

All eight unique trials were presented, once each for practice, before each block of congruent and incongruent trials in each task. Feedback came after each of the practice trials with the word "correct" or "wrong" presented for 0.5 s. Immediately after the practice trials, the participants proceeded to the experimental trials of the block. All eight unique trials were presented twice, giving rise to a total of 16 trials per block and 64 for the whole experiment. To prevent the participants from responding simply to the right or left position as opposed to sweet/sour or angular/round, 8 trials in each block mapped to one response ("sweet"/"sour," angular/round) to the z key, and in the other 8 trials to the m key.

### Analyses

Both accuracy and RTs were analysed as a function of task and congruence. Accuracy and RTs were analysed by means of $2 \times 2$ analysis of variance-type statistics (ATS) with the factors of task and congruence. Note that the ANOVA-type statistic, a robust rank-based nonparametric alternative to the classic ANOVA, is robust to both outliers and the violation of assumptions in classical parametric ANOVA (see *Erceg-Hurn & Mirosevich, 2008*). The analyses were performed in R Statistical Software, as implemented in the {nparLD} package (*Noguchi et al., 2012*). The significant main effects and interactions were further analysed with the Wilcoxon signed-rank test to which Bonferroni corrections were also applied. Effect sizes were also calculated by means of Cliff's Delta.

## Results and discussion
### Accuracy

Data from those participants failing to respond accurately on more than 60% of the trials were excluded from the analyses (a total of 14 participants). Whilst there was a significant main effect of task, $F_{\text{ATS}}(1, \infty) = 15.15$, $p < .001$, the effect of congruence was not significant, $F_{\text{ATS}}(1, \infty) = 0.39$, $p = .530$, nor was the interaction between task and congruence, $F_{\text{ATS}}(1, \infty) = 1.68$, $p = .195$. Wilcoxon signed-rank test revealed that the participants were more accurate in the task in which they had to respond with taste words rather than shapes ($p = .001$, Cliff's Delta $= 0.15$). Figure 7A summarizes the results.

### RTs

Only the trials in which the participants responded correctly were included in the analyses (91.6% of 15,296 trials). The ANOVA-type statistic revealed a significant effect of task, $F_{\text{ATS}}(1, \infty) = 121.31$, $p < .001$, and congruence, $F_{\text{ATS}}(1, \infty) = 13.71$, $p < .001$. The interaction between task and congruence was not significant, $F_{\text{ATS}}(1, \infty) = 3.38$, $p = 0.066$. The participants responded more rapidly in the task in which they responded with taste words rather than with shapes ($p < .001$, Cliff's Delta $= 0.27$). Moreover, participants also responded more rapidly on the congruent than the incongruent trials ($p < .001$, Cliff's Delta $= 0.13$).

The results of Experiment 2 provide evidence for the idea that taste/shape correspondences can indeed produce congruence effects even in the absence of a tastant, but just with shapes and taste words. It is worth mentioning, though, that there was a difference across tasks too: That is, the participants responded more accurately, and more rapidly, in the *taste response task* as compared to the *shape response task*. Moreover, an overall congruence effect was observed across tasks. The results of Experiment 2 extend previous studies assessing taste/shape congruence (e.g., *Fairhurst et al., 2015*; *Liang et al., 2013*) to taste word/shape congruence.

## GENERAL DISCUSSION

Two experiments aimed to assess, first, whether basic taste words and shapes share a semantic space—a set of implicit meanings—that may contribute to the correspondences between these stimuli, and, second, whether these crossmodal correspondences can induce

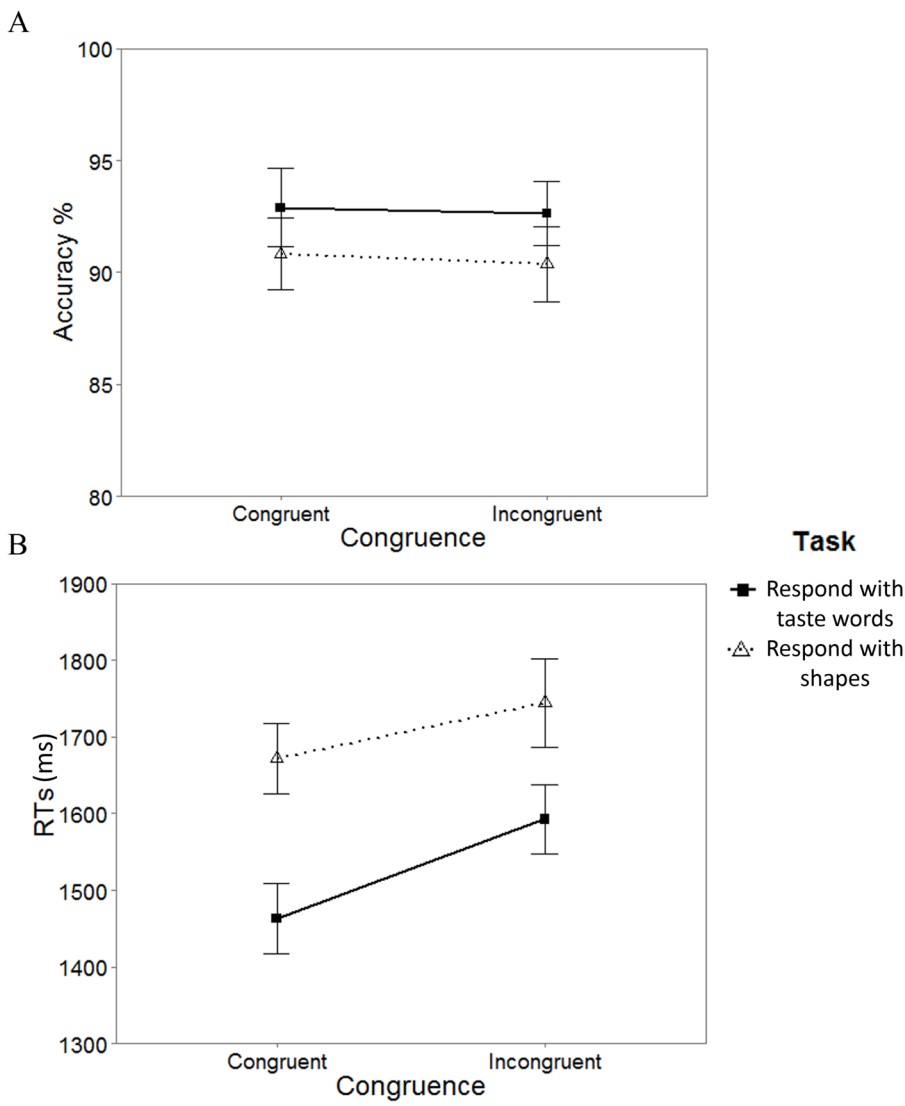

**Figure 7** **Summary of the results of Experiment 2.** (A) Accuracy and (B) Mean reaction times (RTs) in both tasks as a function of congruence. The error bars represent the standard error of the means.

congruence effects when linguistic stimuli rather than tastants are used. Experiment 1 revealed that taste words and shapes can share a semantic space, which is mainly characterized by dimensions related to intensity and hedonic value. Moreover, consistent with previous research (*Spence & Deroy, 2014*, for a review), specific tastes and shapes clustered, "sweet" with round shape and "bitter," "salty," and "sour" with angular shape. A limitation of Experiment 1, though, is that only four shapes were used. Nevertheless, the results are consistent with previous research suggesting that the shapes used are distinctively round and angular (e.g., *Holland & Wertheimer, 1964*; *Lindauer, 1990*; *Lyman, 1979*), that the aforesaid groups of tastes and shapes tend to be rated similarly (*Velasco et al., 2015a*; *Velasco et al., 2016*), and that perceptual dimensions can be described along such dimensions (*Collier, 1996*; *Cavanaugh, MacInnis & Weiss, 2015*).

Experiment 2 introduced a task in which people were instructed to respond either to shapes with taste words or to taste words with shapes, under conditions that defined the stimulus–response relations either congruently (e.g., respond round to "sweet") or incongruently (e.g., respond angular to "sweet"). Both task and congruence mattered: First, the participants were more accurate and faster when responding to taste words with shapes than when responding to shapes with taste words. And second, the participants responded more rapidly with congruent as compared to with incongruent pairings of stimuli and responses.

One important question, relating to Experiment 1, is whether the semantic basis of taste words and shapes may also apply to tastants. Research by *Velasco et al. (2015a)* demonstrated that the ways in which people match taste words and tastants to shapes follow similar patterns (as might be expected).[2] Sweet tends to associate with round shapes whilst bitter, sour, salty, and salty associate with angular shapes. One important direction for future research concerns the evaluation of the semantic space of both taste words and tastants. For example, one could examine the common semantic space for taste words and tastes by running either a semantic differentiation study or a similarity rating experiment on a stimulus set that included both tastes and taste words. Findings of *Gallace, Boschin & Spence (2011)* with different foods, however, do suggest that foods with specific taste qualities aligned in semantic differential scales (see also *Ngo et al., 2013*).

Although the present research focused on taste words and shapes, it is worth considering whether the results would be similar if we had used shape words instead of shapes. Presumably, shape words would operate semantically like shapes *per se*, at least to the extent that taste words operate semantically like tastes (although in both cases there may be some interesting differences between the connotative meanings of perceptual stimuli and the analogous words, as *Osgood, 1960*, suggested with colors and color words). In some instances, words may connote 'prototypes' that cannot easily be realized in particular stimuli. Such a matter may be an interesting direction for future research.

Nearly two decades ago, *Marks (1996)* highlighted that "*The correspondences between primary perceptual meanings and secondary linguistic ones need not be perfect—language and perception do not necessarily carve the world up in precisely the same way (cf. Miller & Johnson-Laird, 1976)—but the connections are nevertheless strong*" (p. 49). The results of Experiment 2 extend previous work on taste/shape associations and taste and shape information processing to taste words and shapes (e.g., *Becker et al., 2011*; *Gal, Wheeler & Shiv, 2007*). As noted before, in the English language, for example, the use of shape-related words such as "sharp" to describe tastes has a long history (*Marks, 1978*; *Williams, 1976*; *Yu, 2003*). This said, even though the effects found in Experiment 2 were small, so too were earlier taste/shape congruence effects reported with perceptual stimuli (*Liang et al., 2013*; note, however, that comparing effect sizes across experimental paradigms is not an easy task given their different nature), and these effects prove noteworthy given the seemingly unrelated nature of basic taste words and shapes.

How to interpret the fact that taste word/shape correspondences gave rise to congruence effects? In order to answer this question, it is important to highlight the fact that the tasks included in Experiment 2 required the participants to learn specific associations (either

[2] This is of particular relevance given the fact that previous research has documented the importance of some non-semantic features of taste words (e.g., typeface features, see *Velasco et al., 2015b*) and/or implicit vocalization, articulation/kinesthesis, and/or sound imagery in conveying meaning (e.g., *Ngo et al., 2013*). For example, one may argue that it is not the word "sweet" but rather its sound symbolic meaning, that guides its matching to round shapes. While we cannot rule out all the specific interactions between the aforesaid variables, it is known that tastants and taste words are similarly matched to shapes varying in terms of their curvature (*Velasco et al., 2015a*).

congruent or incongruent). This said, it is reasonable to assert that there is an implicit relation between specific taste words and shape curvature. In other words, participants may find it easier to respond to a learned stimulus mapping that has a stronger (implicit) association than to one that has a weaker association, thus responding faster to the former. How is such an implicit relationship built? The results of Experiment 1, together with those of *Velasco et al. (2015a)* and *Velasco et al. (2016)*, provide some clues in support of a hedonic association, and preliminary experimental data for a sensory-discriminative association (see also e.g., *Marks, 1978*; *Marks, 2013*; *Parise & Spence, 2013*; *Spence, 2011*, for reviews on possible mechanisms underlying crossmodal correspondences). These ideas are consistent with an affective account of certain associations across the senses (e.g., *Collier, 1996*; *Palmer et al., 2013*). Given that intensity and hedonics are also influenced by other low-level visual properties and shape aesthetic features (e.g., *Palmer, Schloss & Sammartino, 2013*), which influence taste/shape correspondences (*Salgado-Montejo et al., 2015*), it should be reasonable to extend the present results to other visual attributes (e.g., shape symmetry).

The two experiments reported here were conducted online. Even though there is still debate as to the limitations and scope of online research, a number of studies have started to suggest that, in many situations, online research can mimic laboratory results (not to mention the access to larger and more varied samples of participants; see *Woods et al., 2015*, for a review of perception research online). Nonetheless, the differences in hardware, software, and contexts across online participants may certainly introduce some error (though the larger samples likely compensate, see also *Chetverikov & Upravitelev, in press*).

The results of the present study can be of interest to both researchers and practitioners in the context of food and drink design, given that specific stimulus combinations may lead to subtle variations in the processing of taste-related information. Indeed, even though research will be undoubtedly needed, perhaps, by changing the shape of a packaging, plate, or food, it may be possible to influence, for example, the expected and perceived sweetness of foods or drinks without touching their actual concentration (*Velasco et al., 2014*).

To conclude, people appear to respond differently to tastes and shapes when the mappings are consistent rather than inconsistent with the correspondence—an additional piece of evidence to suggest that taste words and shapes share an abstract semantic network and that the existence of crossmodally shared locations in semantic space *ipso facto* define or characterize crossmodal congruence.

### Funding
The authors received no funding for this work.

### Competing Interests
The authors declare there are no competing interests. Andy T. Woods is an employee of Xperiment, UK.

## Author Contributions

- Carlos Velasco conceived and designed the experiments, performed the experiments, analyzed the data, wrote the paper, prepared figures and/or tables, reviewed drafts of the paper.
- Andy T. Woods conceived and designed the experiments, performed the experiments, analyzed the data, wrote the paper, reviewed drafts of the paper.
- Lawrence E. Marks and Charles Spence conceived and designed the experiments, wrote the paper, reviewed drafts of the paper.
- Adrian David Cheok reviewed drafts of the paper.

## Human Ethics

The following information was supplied relating to ethical approvals (i.e., approving body and any reference numbers):

Central University Research Ethics Committee at the University of Oxford (MS-IDREC-C1-2014-056).

## Data Availability

The raw data is included in the Supplemental Information.

## Supplemental Information

Supplemental information for this article can be found online at http://dx.doi.org/10.7717/peerj.1644#supplemental-information.

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
