# Peer review of "The semantic basis of taste-shape associations"

_PeerJ, doi:10.7717/peerj.1644_

## Round 0.1 · original submission · Major Revisions

Our apologies for the delay in this decision.

Please check, and respond to, the comments from the reviewers.

Reviewer 1 ·

Basic reporting

This paper is well written, and easy to follow. It provides sufficient information about the research, and conducts a thoroughly discussion of the results.

Experimental design

This paper has clearly defined the research questions, and the experiment design is reasonable.

Validity of the findings

The data is well controlled, and the conclusion is reliable. My only concern is the number of shapes, only four. I suggest that more shapes would be addressed and tested.

Additional comments

No comments

Reviewer 2 ·

Basic reporting

Both the text and figures of this submission are clear and concise. Only minor typograohical issues to address (e.g. line 314 should read "we proceeded to analyse..." not "analysed").

Experimental design

No suggestions for improvement for this online based experiment where the design is sound with clear well-described research hypotheses. One might consider, in future, limiting the sample to a confined age range (18-35). It would also be useful to know more about specific instructions given to participants.

Validity of the findings

The findings are well presented and clear. The only comment concerns experiment 2 in which despite a lack of a significant interaction further tests were performed to explore the nature of the interaction. This may not be advisable unless the authors believe they have cause to do so.

·

Basic reporting

The authors are reporting their work on the intrinsic association between taste words and visual shape with two experiments. The non-random multisensory (semantic) association is a timely topic of growing interest. The results the authors are reporting are to the point and interesting. Nonetheless, I have several comments and suggestions to improve the manuscript.


[Basic Reporting]

1. Introduction and Discussion should be expanded a little. A good portion of the cited previous works include works of the authors themselves. Despite the fact that the current topic has not been addressed extensively, it should be elaborated a bit more since it can be related to a wide range of important topics.

2. Citation should be reviewed throughout the manuscript.
1-1. It was a bit surprising to find about six articles that are cited in the text body but not listed in the References at the end.
- Spence & Deroy, 2013 (line 55, 336)
- Gal, Wheeler, & Shiv, 2007 (line 60)
- Stewart & Goss, 2013 (line 89)
- Piqueras-Fiszman, Alcaide, Roura, & Spence, 2012 (line 97)
- Miller & Johnson-laird, 1976 (line 365-366)
1-2. There seem to be a few articles listed in the References but not in mentioned in the text body as well.

3. Figure 6 and the text body below the figure don’t match the organization of the figure itself. 6C→A, A→ B, B→ C

4. analysed → analyse (line 314)

5. (line 375) not a sentence.

Experimental design

1. The authors wanted to examine whether bisensory interaction between taste and shape is based on perceptual, semantic, or both processes. To do so, they performed a couple of studies using taste words and visual shape stimuli. For a more complete picture, as the authors themselves acknowledged in discussion, the interaction between taste and shape words, taste and taste words, visual shape and shape words, or even between taste words and shape words should also be examined.

2. In Experiment 1, only two pairs of angular/round shapes were used. They were assessed using adjectives related to valence and intensity. One might question whether the association between “round” shape and the word “sweet” is because of the “roundness” per se as the authors assumed, or other positivity the shape has or induces. For example, the participants might have found the “round” shape looks more attractive. With just a limited number of stimuli, this can be a serious problem. The authors should indicate whether their stimuli are well controlled in other aspects than the critical visual feature (angular/round). Otherwise, discussion over this alternative explanation should at least be included. The number of shape stimuli was larger in Experiment 2, but this question is not irrelevant to the second experiment either.

3. With the block design in Experiment 1, the purpose of the experiment can easily be detected by the participants. Any comments on the potential cognitive penetrability of the current design?

4. In Experiment 2, the association between taste word and shape was given to the participants and the task was speeded classification task. Though the RT results regarding the congruence and the interaction between congruence and task turned out significant, the explicitly learned association might not be optimal to reveal intrinsic association between taste word and shape (raw RTs are pretty long overall). Modified version of IAT(Implicit association task) or Stroop-type task might be worth being considered.

Validity of the findings

1. To examine the validity of the finding from Experiment 1, more details are needed regarding the “hierarchical cluster analysis” method. For example, how the Euclidean distance was considered was not explicitly described in the current manuscript.

·

Basic reporting

The Introduction could use some additional sources besides the authors own work, but otherwise comprehensive and complete.

Experimental design

The procedures could be more clearly explained and technical jargon defined for others who wish to reproduce the findings.

Validity of the findings

The data could have been stronger, but I believe that an outlier procedure could be used to strengthen the results.

Additional comments

Summary:
This study examined the semantic relationship between taste words and round vs pointy shapes (like bouba and kiki stimuli) to see whether they might have similar underlying meanings that associate them together. There were two experiments conducted. The first was a qualitative examination of subjective ratings for taste words and shapes of objects. The results suggested that the taste word “sweet” and “rounded” objects were more likely to be associated, while “bitter, sour, and salty” were associated with “angular” objects. The second experiment was a quantitative response time and accuracy experiment that examined the congruency effects between taste words and shapes of objects. The results showed faster response times and greater accuracy for taste words when matching to shapes compared to shapes when being matched to taste words.


Major Concerns:

I feel like the introduction focused primarily on the work of the authors. For instance, Gallace, Boschin, and Spence did some nice work on the taste of the bouba and kiki effect that might be necessary to mention. What about other crossmodal correspondances? Also, since there were such large samples in each of the experiments (102 in Exp1; 253 in Exp2), I am surprised that the authors did not conduct any gender or age-related effects. For example, the age range in Exp 2 was 18-73 years. One question that comes to mind is, “could experience influence these semantic relationships?” Surely the 73 year old has decades more experience than the 18 years old. On a more fundamental level, should a 73 year old and 18 year old both be included in a data set examining response time? Research has shown that older individuals slow down their response time, therefore potentially dampening any effects found. Indeed there were effects that were borderline reported. Was there an outlier procedure performed? Last, one potential confound might be that the words used could have been verbalized by participants (either aloud or using their inner voice). Might it be possible that the sounds of sweet are round and the sounds of sour are angular?

I have three sort of general things that I believe would strengthen this paper and allow it to have a greater impact:
1) Include more examples. One positive aspect of this topic is that the reader can actually part-take in the study through examples.
2) Conduct more analyses to make the findings stronger. As they stand, the effects are weak, while the conclusion the authors make is strong. In fact, the conclusions might be too strong.
3) Give the reader the significance of this research. I read the paper thoroughly and I am still left with the question of “what does the findings really mean and why should I care?” This question can be asked in many research studies, but it is our job as the experts to lay it out clearly.

Overall, I think the paper needs some work but the experiments are interesting and provide insight into the taste-shape relationship. Perhaps reading some studies on taste-shape synaesthesia could shed some light on these findings or vise versa. I would recommend this paper for publication given the authors consider my suggestions. I do have to admit that I am not familiar with the qualitative stats used in Exp1 and as such cannot accurately evaluate the results.

Minor Comments:

Ln138: When describing the materials, the stimuli presented was confusing to me. I understand the shapes were presented as in Figure 1, but how were the words presented? Were they presented altogether or separately? Were they presented with the shapes? More detail is needed here.

Ln144: The authors might want to quickly define the semantic differential technique. It is technical jargon that readers might not know well.

Ln 250: Figure 6 – What error bars are the authors referring to in this figure? I thought the figure was the stimuli not data, but I could be confused.

Ln 256: When explaining the procedure, I think the authors might have mixed up Figure 6C and 6B? They say, “In one of the tasks (shape response, see Figure 6A), the participants were presented the taste words “sweet” or “sour” (one at a time) and asked to respond with either an angular or round shape (i.e., pairs of shapes taken from Experiment 1, see Figure 6C) – should be 6B. In the other task (taste response, see Figure 6B) – should be 6C, the participants were presented the eight shape stimuli (one at a time) and were asked to response with the taste words “sweet” or “sour”. Again, I could be mistaken.

Ln262: When explain the different congruent and incongruent blocks (which was confusing to read), I am assuming that the blocks were counterbalanced? Or did every participant get congruent and then incongruent each time? I would image that the mental set/response set alone would produce effects on the second task.

Ln288: Forgive my naivety, but what is an “analysis of variance-type statistic?” Why can the authors not perform the standard and widely used ANOVA?

Ln347: the words tastant has appeared numerous times and should be defined in the introduction.

Ln372: the authors mention that the effects are small, but so too are other crossmodal correspondence effects. Perhaps that should be followed up with, “for example, …”
Given that the data was collected online through Mechanical Turk, the authors should provide some limitations to using that type of sample.

---

## Round 0.2 · accepted · Accept

Thanks to your revision and the efforts.

·

Basic reporting

The revised intro and discussion now include more comprehensive reviews of previous literature. With correction of minor issues in the initial submission, the logic of the study is now easy to follow and clear.

Experimental design

Most of the issues raised in the initial review are now mentioned and discussed in the revised manuscript.

Validity of the findings

With revisions, the findings are well presented and clear.

Additional comments

I appreciate the authors' efforts to consider those comments I and other reviewers provided in the first round of review. I'll look forward to seeing more follow-up studies in the future!

·

Basic reporting

In my opinion, the article is in much better shape than the first version. I think the authors have done an excellent job responded to many reviewer comments. The reporting is complete and comprehensive.

Experimental design

I have not had any issues with the design. Still good.

Validity of the findings

The findings are in line with the data and conclusions more accurately reflect that.

Additional comments

I have read the revised manuscript thoroughly and now think that the manuscript is in great shape to be published. I can give my endorsement for publication.